# Position: Certified Correctness in Neural Constraint Reasoning Requires Symbolic Integration

**Shufeng Kong** [1 2]   **Xiaochuan Zhang** [1]   **Caihua Liu** [3 2 4]

## Abstract

Neural solvers for constraint satisfaction problems have achieved remarkable in-distribution accuracy, yet they suffer from a fundamental limitation: persistent constraint violations occur under distribution shifts even when the model reports high confidence. This position paper argues that when hard constraints exist and the cost of verification is relatively low, neural constraint reasoning must prioritize symbolic integration over pure learning. We justify our focus on Sudoku as a representative NP-complete testbed because it exhibits a sharp asymmetry between easy verification and hard solving: checking a candidate solution requires only polynomial time $O(n^2)$, while finding a solution may require exponential search. Through a comprehensive survey of solving methods spanning deterministic algorithms, metaheuristic optimization, learning-based approaches, and language-conditioned reasoning, we demonstrate that neural-only methods without instance-level certification fail to achieve the provable correctness that symbolic and neuro-symbolic approaches provide. We advocate for a bidirectional integration in which neural methods enhance symbolic solvers by learning heuristics and converting percepts into symbols, while symbolic methods verify neural outputs to ensure their reliability. To operationalize this position, we propose a multi-agent certified reasoning framework that demonstrates how this integration can achieve both computational efficiency and provable correctness.

[1]School of Software Engineering, Sun Yat-sen University, Zhuhai, China [2]Department of Computer Science, Cornell University, Ithaca, NY, USA [3]School of Computer Science and Artificial Intelligence, Foshan University, Foshan, China [4]Department of Ecology and Evolutionary Biology, Cornell University, Ithaca, NY, USA. Correspondence to: Caihua Liu <cl2869@cornell.edu>.

*Proceedings of the 43rd International Conference on Machine Learning*, Seoul, South Korea. PMLR 306, 2026. Copyright 2026 by the author(s).

## 1. Introduction

A central debate in modern machine learning concerns the extent to which large-scale statistical learning can replace explicit symbolic reasoning. While neural solvers have achieved remarkable in-distribution accuracy on constraint satisfaction problems (CSPs), they often lack the robustness required for rigorous constraint enforcement. Consider the case of Sudoku: SATNet achieves 98.3% test accuracy on its in-distribution benchmark (Wang et al., 2019), but when the number of given digits shifts from 31–42 to 17–34 (out-of-distribution), accuracy collapses to 3.2% (Miyato et al., 2025), representing a 95.1 percentage point degradation. This is not an isolated failure. Even the leading neural-only solver *without explicit certification*, AKOrN (Miyato et al., 2025), achieves only 89.5±2.5% OOD accuracy despite extensive test-time compute (128 Kuramoto steps and 4,096 samples with energy-based voting). By contrast, the neuro-symbolic system NeurASP achieves 100% constraint satisfaction using only 25 training examples (Yang et al., 2020), requiring orders of magnitude fewer samples than neural-only approaches such as RRN (Palm et al., 2018). These empirical disparities indicate that statistical approximation is fundamentally distinct from logical satisfaction, motivating the central position of this paper.

**Position. Neural constraint reasoning should prioritize symbolic integration over pure learning when: (i) hard constraints exist and are specifiable; (ii) verification costs are low relative to solving; and (iii) violation costs are high.**

This paper focuses on domains where all three conditions hold, including Sudoku and many industrial CSPs such as scheduling, configuration, and compliance checking. In these settings, cheap verification ($O(n^2)$ for Sudoku) can effectively certify expensive-to-produce solutions. We advocate for bidirectional integration: neural methods enhance symbolic solvers by improving accessibility and scalability, while symbolic methods certify neural outputs to ensure trustworthiness.

**Relationship to Prior Work.** Our position shares with Kambhampati et al. (2024) the insight that neural systems

require external symbolic verification, but diverges in a fundamental way. First, we focus on constraint satisfaction problems, where polynomial-time verification (e.g., $O(n^2)$ for Sudoku) contrasts sharply with NP-hard solving, enabling stronger certification claims than planning domains allow. Second, we survey four paradigms (deterministic, metaheuristic, learning-based, and language-conditioned) to demonstrate that the certification gap persists across fundamentally different architectures. Third, whereas LLM-Modulo treats neural components as unidirectional generators verified by symbolic critics, we advocate for bidirectional integration: neural methods enhance symbolic solvers (via learned heuristics and perception), while symbolic methods certify neural outputs—a division of labor that exploits the strengths of both.

**Operational Definition.** To clarify our scope, we employ a taxonomy distinguishing *where* constraint structure is injected and *what* guarantees it yields. We classify systems into three categories: **(i) Purely neural (data-implied):** Systems that (a) invoke no external symbolic computation (such as SAT/SMT/ASP solvers or constraint checkers) at inference time, and (b) do not encode constraints by architectural construction. Constraint satisfaction in these models is statistical rather than certified. **(ii) Architecturally constrained:** Systems where constraints are enforced by design via parameterization, projection, or normalization layers within the network. **(iii) Symbolically integrated:** Systems where constraint semantics are enforced via explicit symbolic verification or solving at inference time, enabling instance-level certification.

**Scope Note: Architectural vs. Symbolic Enforcement.** For continuous constraints (probability simplexes via Softmax; linear inequalities via differentiable QPs (Amos & Kolter, 2017)), architectural enforcement is sufficient. Our position targets the *complementary* regime of discrete combinatorial constraints with global structure (e.g., `all-different` in Sudoku, sub-tour elimination in TSP). Here, end-to-end neural solvers rely on continuous relaxations during training (e.g., SATNet's differentiable SDP of MAXSAT) (Wang et al., 2019): effective soft guidance in-distribution, but the discrete rounding at inference reintroduces violation risk, and this gap cannot be closed by architecture alone without resolving P vs. NP (Garey & Johnson, 1979). Architectural and symbolic enforcement are *complementary*, not competing.

**Why Sudoku? A Controlled "Drosophila" Testbed.** Sudoku is NP-complete (Yato & Seta, 2003) yet exhibits a sharp "easy verification, hard solving" asymmetry: checking takes $O(n^2)$, finding a solution may require exponential search. More importantly, it is a *controlled* testbed: by varying only the clue count ID $[31-42] \rightarrow$ OOD $[17-34]$ while holding all 324 constraints fixed, we isolate failure

modes attributable purely to neural approximation, with no confounding from shifting tasks or constraint sets. Such control is difficult in messier domains (code, scheduling) where many variables change simultaneously. The position generalizes to any domain satisfying (i)–(iii); Section 5.4 confirms this for code generation, hard vehicle routing, and automated theorem proving.

**Contributions.** This paper makes four primary contributions. First, we provide a comprehensive taxonomy of Sudoku solving methods across four paradigms: deterministic algorithms, metaheuristics, neural networks, and large language models (Section 2). Second, we advance three falsifiable empirical claims (Section 3): Claim 3.1 establishes that neural-only methods without explicit certification exhibit >10% violation rates under distribution shift; Claim 3.2 demonstrates that test-time scaling yields diminishing returns and fails to distinguish correct from incorrect outputs; and Claim 3.3 shows that neuro-symbolic approaches achieve dramatic sample efficiency gains. Third, we articulate **strategies for bidirectional integration**, demonstrating how neural methods can enhance symbolic solvers (via perception and heuristics) and how symbolic methods must certify neural outputs to ensure trustworthiness (Section 4). Fourth, we propose the **Proposer-Verifier-Solver (PVS) framework**, a concrete multi-agent architecture that operationalizes these strategies to achieve both computational efficiency and provable correctness (Section 5).

**Falsifiability.** To ensure scientific rigor, we define a strict refutation criterion: If a *purely neural (data-implied)* system achieves a <1% violation rate on a preregistered OOD benchmark *without* (i) invoking symbolic solvers or explicit constraint checkers at inference time, and *without* (ii) enforcing the task-defining constraints in $\mathcal{C}$ by architectural construction, then this thesis is refuted.

**Conflict of Interest Disclosure.** The authors declare no financial conflicts of interest: no commercial product is evaluated, and no author is employed by an organization that produces the systems compared in this paper.

## 2. A Taxonomy of Constraint Satisfaction Paradigms

To locate the precise failure mode of modern solvers, we categorize Sudoku solving methods into four paradigms: deterministic algorithms, metaheuristic optimization, end-to-end neural learning, and language-conditioned reasoning. We evaluate each paradigm against three criteria central to our position: **flexibility** (handling unstructured inputs), **efficiency** (inference latency), and **certified correctness** (guaranteed satisfaction). Table 1 summarizes these trade-offs, identifying a critical "Certification Gap"

in modern learning-based approaches.

## 2.1. Deterministic Algorithms: The Baseline of Certifiability

Deterministic methods—including Dancing Links (DLX) (Knuth, 2000), SAT solvers (Biere et al., 2020), and constraint propagation (Norvig, 2006)—represent the gold standard for correctness.

Modern implementations such as Tdoku (Dillon, 2019) leverage SIMD optimizations (AVX-512) to achieve microsecond-level solve times ($\sim$2.7 $\mu$s/puzzle). Crucially, these methods provide **correctness by construction**: they do not return a result unless it provably satisfies all constraints. Their limitation is not reliability, but rigidity; they cannot consume the unstructured inputs (images, natural language) characteristic of real-world AI deployment.

## 2.2. Metaheuristic Optimization: Search Without Guarantees

Metaheuristics—such as simulated annealing (Lewis, 2007) and genetic algorithms (Mantere & Koljonen, 2007)—recast constraint satisfaction as an energy minimization problem. While more flexible than exact solvers, they suffer from phase transitions. Lewis (2007) showed that on order-5 Sudokus ($25 \times 25$), success rates can drop to 30% near the critical hardness threshold. Unlike deterministic methods, metaheuristics provide no guarantee of convergence; they may stagnate in local optima where constraints remain violated, making them unsuitable for safety-critical certification.

## 2.3. End-to-End Neural Learning: The Statistical Trap

This paradigm attempts to learn constraint satisfaction from data, treating logical necessities as statistical regularities. Architectures range from Recurrent Relational Networks (RRN) (Palm et al., 2018) to differentiable solvers like SATNet (Wang et al., 2019). While these models achieve high in-distribution (ID) accuracy (e.g., 98.3% for SAT-Net), they exhibit a fundamental **lack of robustness**. Under distribution shift (AKOrN split), SATNet's accuracy collapses to 3.2% (Miyato et al., 2025). Even AKOrN (Miyato et al., 2025), which integrates oscillator dynamics for better stability, relies on energy-based voting to achieve 89.5% OOD accuracy. Because constraints are soft training signals rather than hard inference gates, these methods inevitably produce "almost correct" solutions that are logically invalid.

**The Neuro-Symbolic Exception(Control Baseline).** Systems like NeurASP (Yang et al., 2020) deviate from this pattern by delegating the solving step to a symbolic ASP backend. This isolation of responsibilities allows NeurASP

to achieve 100% constraint satisfaction given perceived inputs. The failure mode shifts from *reasoning* (solving the puzzle) to *perception* (reading the digits), proving that integration can preserve certification where pure learning fails. We treat NeurASP as a *control baseline* in this paper: its neural component performs only perception (mapping digit images to label distributions), and the ASP solver constructs the full solution. The neural network does not generate candidate solutions; the system therefore isolates the effect of explicit symbolic specification from any contribution of neural search. The Proposer-Verifier-Solver framework introduced in Section 5 lies at the opposite end of the spectrum: the neural network actively proposes candidates and the symbolic component certifies them.

## 2.4. Language-Conditioned Reasoning: The Illusion of Logic

Large Language Models (LLMs) represent the newest frontier, attempting to solve CSPs via token-by-token reasoning (Chain-of-Thought). Despite improvements in "System 2" reasoning models (e.g., OpenAI's o1), pure LLMs struggle with strict global constraints. On Sudoku-Bench (Seely et al., 2025), GPT-5 achieves only 33% accuracy on the `challenge_100` set.

The core issue is that autoregressive generation is probabilistic; an LLM can "reason" its way to a constraint violation with high confidence. However, when LLMs are augmented with tool use (LLM-Modulo (Kambhampati et al., 2024)), effectively becoming neuro-symbolic controllers, they can regain correctness—but only if the symbolic tool is treated as the source of truth.

## 2.5. Summary: The Certification Gap

This survey reveals a distinct "Certification Gap". We have methods that are fast and certified but rigid (Deterministic), and methods that are flexible and general but uncertified (Neural/LLM). Purely neural approaches—regardless of scale—cannot bridge this gap because they approximate discrete validity with continuous probability. This necessitates the **bidirectional integration** proposed in this paper: using neural models for flexible perception and heuristics, while retaining symbolic engines for the final, non-negotiable certification of correctness.

## 3. Empirical Evidence: The Limits of Statistical Learning

Neural-only methods without explicit symbolic certification can achieve near-perfect in-distribution accuracy but degrade substantially under distribution shift. By contrast, neuro-symbolic methods maintain constraint satisfaction by delegating to symbolic solvers.

*Table 1.* The Certification Gap. While deterministic methods guarantee correctness, they lack the flexibility to handle raw perceptual inputs. Neural and LLM approaches offer flexibility but sacrifice certification, leading to OOD failures. Neuro-symbolic integration (the advocated position) bridges this gap.

| Paradigm | Input Modality | Typical Speed | Certified? | Failure Mode |
|---|---|---|---|---|
| Deterministic | Structured | $\mu$s–ms | **Yes** | Rigid input requirements |
| Metaheuristic | Structured | ms–s | No | Local optima convergence |
| End-to-End Neural | Raw/Structured | ms–s | No | **OOD Degradation** |
| Language-Cond. | Text/Multimodal | s–min | No | Hallucinated reasoning |

We advance three empirical claims supported by recent theoretical findings:

- **Claim 1** (OOD Violations): State-of-the-art neural-only methods *without explicit symbolic certification* exhibit residual constraint violations under distribution shift that further test-time compute cannot eliminate.

- **Claim 2** (Compute $\neq$ Certificates): Test-time scaling improves average accuracy but provides no mechanism to distinguish correct from incorrect individual outputs, leaving a "validity gap" that compute alone cannot close.

- **Claim 3** (Integration Efficiency): Neuro-symbolic approaches achieve orders-of-magnitude sample efficiency gains by removing constraint satisfaction from the hypothesis space.

### 3.1. Persistent OOD Violations

**Claim 1.** Even state-of-the-art neural-only methods *without explicit symbolic certification* exhibit residual constraint violations under distribution shift that further test-time compute cannot eliminate.

Figure 1 visualizes the performance gap between in-distribution and out-of-distribution settings. The pattern is striking: neural solvers that appear near-perfect on in-distribution data suffer severe degradation when the difficulty distribution shifts.

This degradation is not merely an engineering failure but a theoretical inevitability. As noted by Balestriero et al. (2021), neural inference in high-dimensional spaces almost always involves extrapolation beyond the training manifold. Models like SATNet, even when incorporating differentiable relaxations of constraint structure, fail to generalize these structures robustly. Increasing test-time compute, as seen in ConsFormer and AKOrN, partially mitigates the problem but cannot eliminate it. Concretely, AKOrN plateaus at $\sim$10.5% violations on our preregistered Sudoku OOD protocol even at its maximum compute budget (128 Kuramoto steps, 4,096 samples, energy-based voting). Even with extensive sampling and voting, neural-only methods leave a persistent residual of constraint violations.

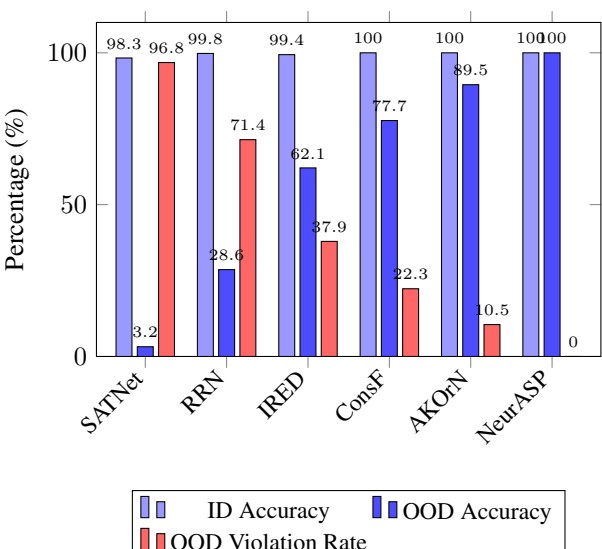

*Figure 1.* Performance degradation under distribution shift. All methods achieve near-perfect in-distribution (ID) accuracy, but neural-only methods *without instance-level certification* show substantial OOD degradation (red bars). NeurASP maintains 0% violation rate by delegating to a symbolic solver.

NeurASP achieves 100% constraint satisfaction (given perceived inputs) through a categorically different mechanism: it delegates constraint satisfaction entirely to an ASP solver. The neural component handles only perception (digit recognition), while the symbolic component guarantees correctness. This architectural division of labor eliminates the certification gap by construction.

### 3.2. Compute Cannot Produce Certificates

**Claim 2.** Test-time scaling yields diminishing returns and provides no mechanism to distinguish correct from incorrect outputs.

A natural response to OOD degradation is to increase test-time compute. ConsFormer's scaling curve illustrates the limitations of this approach: moving from 2K to 10K iterations (a $5\times$ increase in compute) yields only $\sim$12 percentage points of improvement, plateauing well below 100%. Similarly, AKOrN requires 4,096 forward passes per puz-

zle, each involving 128 internal Kuramoto iteration steps, yet still fails on $\sim 10.5\%$ of instances. By contrast, Tdoku solves the hardest puzzles in $\sim 42\mu$s. The compute ratio exceeds $10^8$, yet the neural approach provides no correctness guarantee.

The deeper issue is epistemic rather than computational. Xu et al. (2024) prove that for any computable function approximated by a neural network, there exist inputs where the model produces incorrect outputs with non-negligible probability. Neural confidence scores are notoriously miscalibrated (Guo et al., 2017); thus, a high confidence score is not a certificate of correctness. A system that is "usually right" is categorically different from one that is "provably right."

**The Aggregate-Instance Gap.** Accuracy is an aggregate property measured over distributions, whereas certification is an instance property required for each output. Even 99.9% expected accuracy leaves the question "is this specific output correct?" unanswerable from the model alone. Symbolic verification supplies exactly this missing per-instance information: a binary signal (valid/invalid) that is perfectly calibrated by construction.

### 3.3. Integration Enables Sample Efficiency

**Claim 3.** Neuro-symbolic approaches achieve substantial sample efficiency gains by delegating constraint satisfaction to symbolic solvers.

The resulting sample-efficiency gap is striking: NeurASP achieves 100% constraint satisfaction using only 25 training examples (Yang et al., 2020), whereas RRN requires 216,000 examples to reach comparable in-distribution performance (Palm et al., 2018).

This contrast follows directly from the "Learn vs. Specify" principle. In NeurASP, the neural component is tasked primarily with perception (mapping pixels to digits), while constraint reasoning is enforced by predefined ASP rules. In RRN, the network must implicitly learn both the visual representation and the logical rules of Sudoku from data. The orders-of-magnitude difference in sample efficiency is a predictable consequence of removing constraint satisfaction from the hypothesis space.

One might object that comparing NeurASP and RRN is unfair because they do not instantiate identical learning problems. We view this mismatch not as a confounding variable, but as the *mechanism of action*. The fact that neuro-symbolic architectures allow us to simplify the learning problem—converting a reasoning task into a perception task—is precisely why they are superior for constrained domains.

Claim 3 measures *paradigm cost*, not the relative merit

of two specific learning algorithms; the independent variable is the presence vs. absence of explicit symbolic specification. RRN tackles the same Sudoku task with constraints *learned* from data rather than specified: it consumes 216,000 training examples ($\sim 8,640\times$ NeurASP's 25) and *still* collapses under the same OOD shift. The certification gap is therefore *architectural*, not a matter of dataset size: a soft-regularizer architecture obtains its constraint signal from continuous relaxations of discrete constraints, and those relaxations hold only in expectation—no amount of additional data converts them into the instance-level guarantee that a symbolic verifier provides by construction.

## 4. Strategies for Bidirectional Integration

The case for bidirectional integration rests on a fundamental computational asymmetry: for NP-complete problems, **solving is hard (exponential), but verification is cheap (polynomial).** A SAT formula may require exponential search to satisfy, yet any proposed assignment can be checked in linear time. This asymmetry suggests an optimal division of labor: neural networks excel at efficient generation through pattern recognition (System 1), while symbolic systems excel at rigorous verification through constraint checking (System 2).

We therefore advocate for a bidirectional strategy where each component compensates for the other's deficits.

### 4.1. Neural Methods Enhancing Symbolic Solvers

Symbolic solvers guarantee correctness but face two practical barriers: they struggle with unstructured inputs (the *symbol grounding problem*), and generic heuristics may be suboptimal for specific data distributions. Neural methods address both.

**Neural Perception (Symbol Grounding).** Real-world constraints often operate on raw modalities like images or text, which symbolic solvers cannot ingest. Neural encoders bridge this gap by mapping raw perception to discrete symbols. NeurASP (Yang et al., 2020) exemplifies this decomposition: a CNN grounds digit images to probability distributions, while an ASP solver enforces Sudoku constraints over these distributions. This separation allows the system to isolate *perception errors* (misreading a digit) from *reasoning errors* (violating a constraint). Crucially, while the neural component provides the *specification*, the symbolic component ensures the *satisfaction* of that specification.

**Neural Heuristics (Learned Search).** For hard instances, the bottleneck is the search space. Deterministic solvers typically rely on generic heuristics (e.g., VSIDS). Neural networks can learn instance-specific distributions to

guide this search, acting as a "learned intuition" that prunes the search space without compromising soundness. Approaches like Graph-Q-SAT (Kurin et al., 2020) and AlphaGo's MCTS (Silver et al., 2016) demonstrate that learned priors can speed up solving by orders of magnitude. The key property is that distribution shift degrades only *efficiency*, not *correctness*.

### 4.2. Symbolic Methods Certifying Neural Outputs

The converse direction is the primary focus of this paper: ensuring trustworthiness. As Kambhampati et al. (2024) argue, autoregressive models "cannot autonomously self-verify" because they apply to verification the same learned heuristics that produced the potentially flawed output. Self-verification is circular; external verification is foundational. We formalize this strategy as **Verification-Interposed Execution**. The system output $\hat{y}$ is defined not as the direct output of a neural function $f_\theta$, but as the conditional result of a gatekeeping function:

$$\hat{y} = \begin{cases} f_\theta(x) & \text{if } \text{VERIFY}(f_\theta(x), \mathcal{C}) = \texttt{True} \\ g_{\text{sym}}(x) & \text{otherwise} \end{cases} \quad (1)$$

where $g_{\text{sym}}$ is a sound symbolic solver (the fallback). For Sudoku, verification takes less than $1\mu s$ ($O(n^2)$), which is negligible compared to the $\sim 10$ms required for neural inference. With AKOrN's 89.5% OOD accuracy, this strategy allows 89.5% of queries to use the fast neural path, while the remaining 10.5% trigger the solver—ensuring 100% certification with minimal amortized overhead.

### 4.3. The Certification Invariant and Inversion of Control

These strategies culminate in a single architectural invariant. **The Certification Invariant:** A reasoning system is *certified* if and only if no output reaches the user without passing a symbolic constraint check. When hard constraints exist, verification is cheap, and violations are costly, this distinction is not a matter of degree—it is categorical. It separates systems that are "usually correct" from those that are "provably correct."

**Inversion of Control vs. Prior Paradigms.** The novelty here is not combining neural and symbolic computation, but the *architectural* insistence that the symbolic verifier be a mandatory gatekeeper at the system level rather than an optional aid (Table 2). Paradigm A softens discrete constraints into differentiable surrogates that hold only in expectation; discrete rounding at inference reintroduces violation risk. Paradigm B is the dominant LLM-agent pattern, but Kambhampati et al. (2024) document two systematic failure modes—*invocation hallucination* (the model emits an answer instead of calling the tool) and *output override*

*Table 2.* Three paradigms of neuro-symbolic integration; only Inversion of Control places a sound verifier on every output path.

| Paradigm | Examples | Failure / guarantee |
|---|---|---|
| A. Soft symbolic (regularizer) | SATNet (Wang et al., 2019), Scallop (Huang et al., 2021) | ID 98.3% →OOD 3.2% (Miyato et al., 2025); silent failure, no fallback |
| B. Ad-hoc tool-use | LLM-Modulo (Kambhampati et al., 2024), Toolformer (Schick et al., 2023), ReAct (Yao et al., 2023b) | Invocation hallucination + output override leave correctness probabilistic |
| C. **Inversion of Control** | PVS (this paper), ICF, BFS-Prover-V2 | Every output traverses $V$; Prop. 5.1 gives a structural guarantee |

(the model discards a correct symbolic result). PVS enforces an *Inversion of Control*: the verifier is a system-level gatekeeper, and the proposer is architecturally precluded from bypassing or overriding it. Proposition 5.1 states the resulting structural—rather than probabilistic—zero-violation guarantee.

## 5. The Proposer-Verifier-Solver (PVS) Framework

To operationalize the strategies in Section 4, we propose the **Proposer-Verifier-Solver (PVS)** framework. This architecture fulfills the "multi-agent" proposition of our abstract by modeling constraint satisfaction as a collaborative control loop between a neural agent (optimized for efficiency) and a symbolic environment (optimized for correctness).

### 5.1. Architectural Components

The framework $\mathcal{F}$ decomposes reasoning into three functional modules:

1. **The Neural Proposer** ($P_\theta$): A learnable statistical model (e.g., Transformer, GNN) mapping inputs $x$ to candidate assignments $\hat{y}$. Functioning as the "System 1" agent, it prioritizes high-throughput generation and handles unstructured modalities but offers no correctness guarantees.

2. **The Symbolic Verifier** ($V$): A deterministic polynomial-time function that checks if $\hat{y} \models \mathcal{C}$. Crucially, $V$ returns structured diagnostics $D$ (e.g., specific constraint violations) rather than a simple boolean, enabling targeted refinement.

3. **The Symbolic Solver** ($S$): A complete solver (e.g., Tdoku, SAT, ASP) acting as the fallback mechanism.

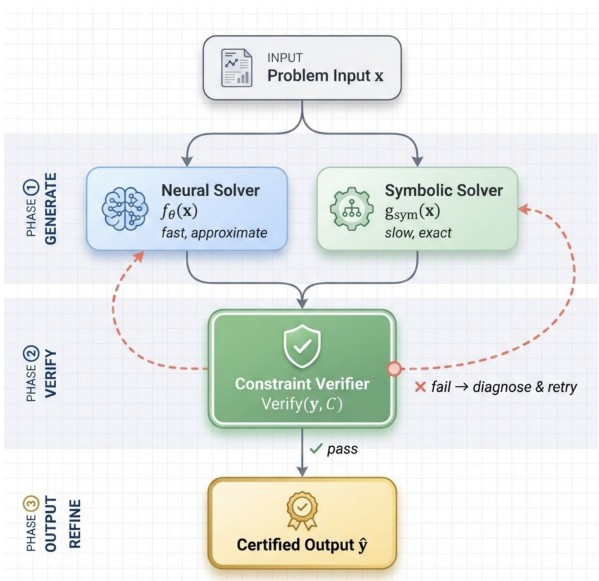

*Figure 2.* The Proposer-Verifier-Solver (PVS) Architecture. The neural agent generates candidates; the symbolic verifier acts as a gatekeeper. Failed candidates trigger diagnostic feedback for refinement. If refinement fails, the symbolic solver serves as the safety net, ensuring the Certification Invariant is never violated.

It guarantees that a valid solution $y^*$ or a proof of unsatisfiability is always attainable.

### 5.2. The Multi-Agent Reasoning Loop

Standard neural inference is a feed-forward process. We restructure this as an agentic refinement loop (Algorithm 1): the Verifier produces structured diagnostics on each failed candidate, and an operator CONSTRUCTFEEDBACK (defined below) folds those diagnostics into the proposer's conditioning input *without updating any parameters*.

---

**Algorithm 1** The PVS Agentic Control Loop

---

1: **Input:** Observation $x$, Constraints $\mathcal{C}$, Budget $T$
2: $c_0 \leftarrow x$; $\hat{y}_0 \leftarrow P_\theta(c_0)$ {Initial proposal (System 1)}
3: **for** $t = 0$ **to** $T$ **do**
4:    valid, $D_t \leftarrow V(\hat{y}_t, \mathcal{C})$ {$D_t$: localized diagnostics}
5:    **if** valid **then**
6:       **return** $\hat{y}_t$ {Certified Fast Path}
7:    **end if**
8:    $c_{t+1} \leftarrow$ CONSTRUCTFEEDBACK$(x, \hat{y}_t, D_t)$ {inject $D_t$ into conditioning}
9:    $\hat{y}_{t+1} \leftarrow P_\theta(c_{t+1})$ {Refinement (System 2)}
10: **end for**
11: **return** $S(x)$ {Certified Safe Path (Fallback)}

---

The loop implements **Test-Time Adaptation**: only the conditioning input $c_t$ changes across iterations; the proposer's parameters $\theta$ are never updated. Unlike scalar loss

gradients, the diagnostics $D_t$ returned by $V$ provide semantic, spatially localized grounding (e.g., "cell (3,4) conflicts with row 3").

**ConstructFeedback: a cross-space conditioning operator.** CONSTRUCTFEEDBACK maps symbolic diagnostics into a conditioning signal in the proposer's native representation space. Three canonical realizations span the dominant proposer families: (i) *LLMs*: diagnostics are prepended as execution context, and in-context learning (Brown et al., 2020) reshapes attention so the model corrects its output from verifier feedback without fine-tuning (Olausson et al., 2024). (ii) *GNNs / message-passing*: diagnostics become edge masks or penalty flags on violating nodes, conditioning the next forward pass on a modified graph. (iii) *Energy-based / oscillator dynamics*: diagnostics enter an additive symbolic-energy term $E_{\text{sym}}(y)$ that steers inference-time sampling (Du & Mordatch, 2019); AKOrN's Kuramoto oscillator dynamics with energy-based voting (Section 3) instantiates this family.

**Spatially localized diagnostics, not global traces.** PVS is not classical guess-and-check (which discards and restarts) nor CEGAR (Clarke et al., 2000) (which refines a global abstraction system-wide). Diagnostics are scoped to the current candidate: "cells (3,4) and (3,7) share value 5; row-3 all-different violated" (Sudoku); "file, line, column, AST node, violation type" (code); UNSAT cores (SMT). Iterative refinement decreases fallback rate across rounds (CaR drives TSPTW-100 infeasibility from 38.22% to 0.03% within $T_R$=10 steps).

### 5.3. Theoretical Guarantee: Asymmetric Trust

The central property of this framework is **Asymmetric Trust**. We place zero trust in the neural component's reliability, yet the system achieves a provable guarantee of correctness.

**Proposition 5.1** (Certification Preservation). *For any input $x$ and constraint set $\mathcal{C}$, if the PVS framework returns an output $y$, then $y \models \mathcal{C}$ (or $y$ is a verified proof of unsatisfiability).*

Algorithm 1 has exactly two exit paths: Line 5 returns $\hat{y}_t$ only after $V$ certifies it, and Line 10 falls back to the sound, complete solver $S$. Since no execution path bypasses a sound symbolic check, the system's violation rate is structurally 0%, independent of $P_\theta$'s reliability.

### 5.4. Cross-Domain Generalization Beyond Sudoku

Table 3 summarizes three high-stakes domains where the propose–verify–refine pattern is now the leading approach.

*Code generation* (verifier: compiler / unit tests): on HumanEval Python, Reflexion's verbal-feedback loop with

*Table 3.* Cross-domain evidence for PVS-style pipelines. "Final" is the certified rate after fallback/refinement.

| Domain (verifier) | Neural-only failure | Fallback / refine | | Final |
|---|---|---|---|---|
| Sudoku OOD (checker; ours) | 10.5% (AKOrN, max compute) | 10.5% solver | → | 100% |
| Code gen, HumanEval (compiler / tests) | 20% (GPT-4 zero-shot pass@1) | verbal critic + test feedback | | 91% (Reflexion) |
| Hard VRP, TSPTW-100 (feas. check) | 38.22% (POMO) | ~38% 0.03% | → | 99.97% |
| Thm. proving, miniF2F (Lean) | ~47% (best ~53% pass@1) | failed → BFS backtrack | | 95.08% |

self-generated tests improves GPT-4 pass@1 from 80% to 91% (Shinn et al., 2023); failed candidates trigger a reflection-conditioned regeneration instead of being silently accepted. *Hard vehicle routing* (feasibility checker): on TSPTW-100, AM/POMO produces 38.22% infeasible routes (Bi et al., 2026); PIP (Bi et al., 2024) reduces this to 6.96% and CaR (Bi et al., 2026) to 0.03% within ten refinement steps. *Automated theorem proving* (Lean kernel): best single-pass Kimina-Prover (Wang et al., 2025) reaches ~53% pass@1; with the Lean kernel as mandatory verifier at every BFS node, BFS-Prover-V2 reaches 95.08% (Xin et al., 2025). In all four settings, the same architectural pattern—fixed neural proposer, mandatory symbolic verifier with structured diagnostics, symbolic fallback—separates probabilistic from certified correctness.

# 6. Alternative Views

## 6.1. Objection 1: "Scale Will Eventually Solve It"

**Steelman Argument.** The history of deep learning is a graveyard of symbolic objections. Domains once thought to require hard-coded priors—from protein folding to code generation—were revolutionized by general-purpose scaling. Recent reasoning models demonstrate emergent capabilities like self-correction and long-horizon planning. It is reasonable to hypothesize that a sufficiently large neural solver, trained on vast datasets of constraints (e.g., generated by solvers), effectively internalizes the logic of verification, closing the certification gap through sheer capacity rather than architectural hybridization.

**Response.** We regard this as the strongest objection, yet it fails on three grounds: empirical, theoretical, and economic. First, empirical evidence contradicts the "emergence" hypothesis for strict constraints. On SUDOKU-

BENCH (Seely et al., 2025), a benchmark explicitly designed to test creative constraint reasoning, even frontier models like GPT-5 achieve only 33% accuracy on the challenge set. Specialized neural solvers like AKOrN (Miyato et al., 2025) and ConsFormer (Xu et al., 2025) plateau well above the <1% violation threshold despite massive test-time compute. Scale improves average-case intuition but does not appear to yield the worst-case guarantees required for certification.

Second, theoretical limitations suggest this is a category error. Balestriero et al. (2021) prove that test predictions in high dimensions are geometrically equivalent to extrapolation. Furthermore, Xu et al. (2024) establish that for any computable function approximated by a neural network, there exist inputs where the model produces incorrect outputs. "Self-verification" does not solve this; it merely applies the same approximate heuristics to the output that generated it (circular certification).

Third, the economic argument favors integration. Even if scaling *could* achieve 99.99% satisfaction, a symbolic verifier (running in $< 1\mu s$) combined with a fallback solver achieves 100% correctness at a fraction of the inference cost. Using a trillion-parameter model to emulate a linear-time consistency check is a massive allocation inefficiency.

## 6.2. Objection 2: "Tool Use Is Sufficient"

**Steelman Argument.** We do not need intimate neuro-symbolic integration because LLMs can simply invoke external tools (Schick et al., 2023). An LLM that generates Python code to call a Z3 solver achieves perfect accuracy without modifying the model architecture. Therefore, flexible "agentic" tool use supersedes the need for the specialized PVS frameworks proposed here.

**Response.** We partially agree: tool-augmented LLMs are a valid instance of symbolic integration. However, current implementations are *unprincipled*. The gap between "merely calling tools" and "reliably calling correct tools with correct inputs" is substantial. As noted by Kambhampati et al. (2024), frontier agents exhibit three systematic failure modes: invocation hallucination (the model emits an answer instead of calling the tool), formulation errors (translating the problem incorrectly into the solver's language), and output override (ignoring the solver's result when it contradicts the model's prior). Our position is not against tool use but against *ad-hoc* tool use. We advocate for **architectural enforcement**: the tool invocation must be triggered by formal conditions, and the solver's output must be treated as authoritativerather than as just another context token. Moreover, the verifier is the *trust boundary*: every output crosses it, and the proposer can neither bypass nor override it—the Inversion of Control absent from Paradigm B (Table 2).

### 6.3. Objection 3: "Real-World Constraints Are Too Messy"

**Steelman Argument.** Sudoku features perfectly specified, complete, and static constraints. Real-world CSPs (e.g., supply chain logistics, legal compliance) involve partially known constraints, soft constraints with violation costs, or logically infeasible instances. Focusing on "NP-complete" exact satisfaction is an academic idealization that does not transfer to the noisy reality of deployment.

**Response.** While Sudoku is an idealized proxy, the principle of symbolic integration degrades gracefully in messy environments:

- **Partially Known Constraints:** Integration allows for a hybrid state. In drug discovery, known physics (e.g., valence rules) are enforced symbolically, while unknown properties (e.g., toxicity) are approximated neurally. This yields *partial certification*—strictly better than zero certification.

- **Soft Constraints:** The framework adapts by shifting from "check validity" to "compute cost." Symbolic solvers can compute exact penalty values for soft constraints, providing a principled loss signal rather than a learned approximation of the loss.

- **Infeasible Instances:** When a problem has no solution, a neural model often hallucinates a "best guess." A symbolic solver returns `UNSAT` with a proof or a Minimal Unsatisfiable Core. This diagnostic value is unique to symbolic reasoning and critical for human decision support.

**Tiered verification and amortized cost.** Where industrial verification itself is expensive (e.g., SMT-based program checking), our position rests on *relative asymmetry*, not absolute cheapness: for any NP problem, verification is asymptotically cheaper than exhaustive solving. Practice further amortizes cost via multi-tier verification—syntax/type checks before bounded model checking before full SMT (Newcombe et al., 2015; Brooker & Desai, 2024). In our experiment, 89.5% of candidates pass the $O(n^2)$ checker before any solver call, giving amortized cost $0.895 \cdot O(n^2) + 0.105 \cdot O(\text{solver})$ per query. When verification times out, PVS falls back to a safe template; partial certification is strictly safer than zero.

**Verifier complexity across NP and beyond.** Sudoku's $O(n^2)$ verifier is heavier than that of most canonical NP-complete problems (3-SAT: $O(n)$; Hamiltonian cycle: $O(n)$; graph $k$-coloring: $O(|E|)$); the asymmetric regime PVS exploits is the norm, not a Sudoku artifact. Beyond NP (PSPACE/EXPTIME) exact verification may not be polynomial, but PVS retains value via cheap falsification and bounded partial verification.

### 6.4. Objection 4: "Autoformalization and Neural Verifiers Suffice"

**Steelman Argument.** A growing line of work uses LLMs as *verifiers* or *autoformalizers*: in theorem proving, an LLM can translate a natural-language proposal into Lean and check it, or judge the proof directly. If the autoformalizer or neural judge is strong enough, an explicit symbolic verifier may be unnecessary.

**Response.** Autoformalization is crucial for domains whose constraints are not yet machine-checkable, but it does not displace symbolic certification. First, when an autoformalizer feeds a *sound* symbolic kernel (Lean, Z3, Tdoku), the system is an instance of PVS: the proposer becomes "LLM + autoformalizer," but the gatekeeper is still a sound verifier; the Certification Invariant holds, and Xin et al. (2025) reach 95.08% on miniF2F precisely because the Lean kernel certifies every accepted step. Second, when the verifier is itself a learned neural judge, soundness becomes probabilistic and we are back in Paradigm B of Table 2; even strong learned verifiers exhibit false-accept rates incompatible with the <1% violation threshold we adopt. Third, autoformalization is itself consistent with our call for bidirectional integration—neural translation *into* symbolic representations, followed by symbolic certification—and so expands the set of domains for which PVS applies rather than supplanting the soundness requirement on the final verifier.

## 7. Conclusion

We have defended a falsifiable position: when hard constraints are explicit, verification is cheap, and violations are costly, neural constraint reasoning must prioritize symbolic integration over pure learning. Neural-only solvers exhibit persistent OOD violations that further test-time compute does not eliminate; "usually right" is epistemically distinct from "provably right"; and neuro-symbolic systems achieve orders-of-magnitude gains in sample efficiency by delegating logic to solvers. Section 5.4 confirms the same pattern across code generation, hard vehicle routing, and theorem proving.

**Call to Action.** We invite benchmark designers to consider reporting Violation Rate alongside accuracy, together with per-instance certification metadata that records whether each output was verified, produced by symbolic fallback, or left uncertified. We encourage system designers to explore the PVS pattern and suggest that certification status be regarded as a meaningful disclosure for "reasoning" claims on CSPs. We would warmly welcome attempts to refute the criterion stated in Section 1; such a refutation would mark a genuine breakthrough in the capacity of statistical learning to approximate logic. Until then, we hope the community will continue building systems that are *provably* right, not merely usually so.

## Acknowledgements

This work was supported by the National Natural Science Foundation of China (Grant No. 62506090) and the National Key R&D Program of China (Grant No. 2025YFF0523900).

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

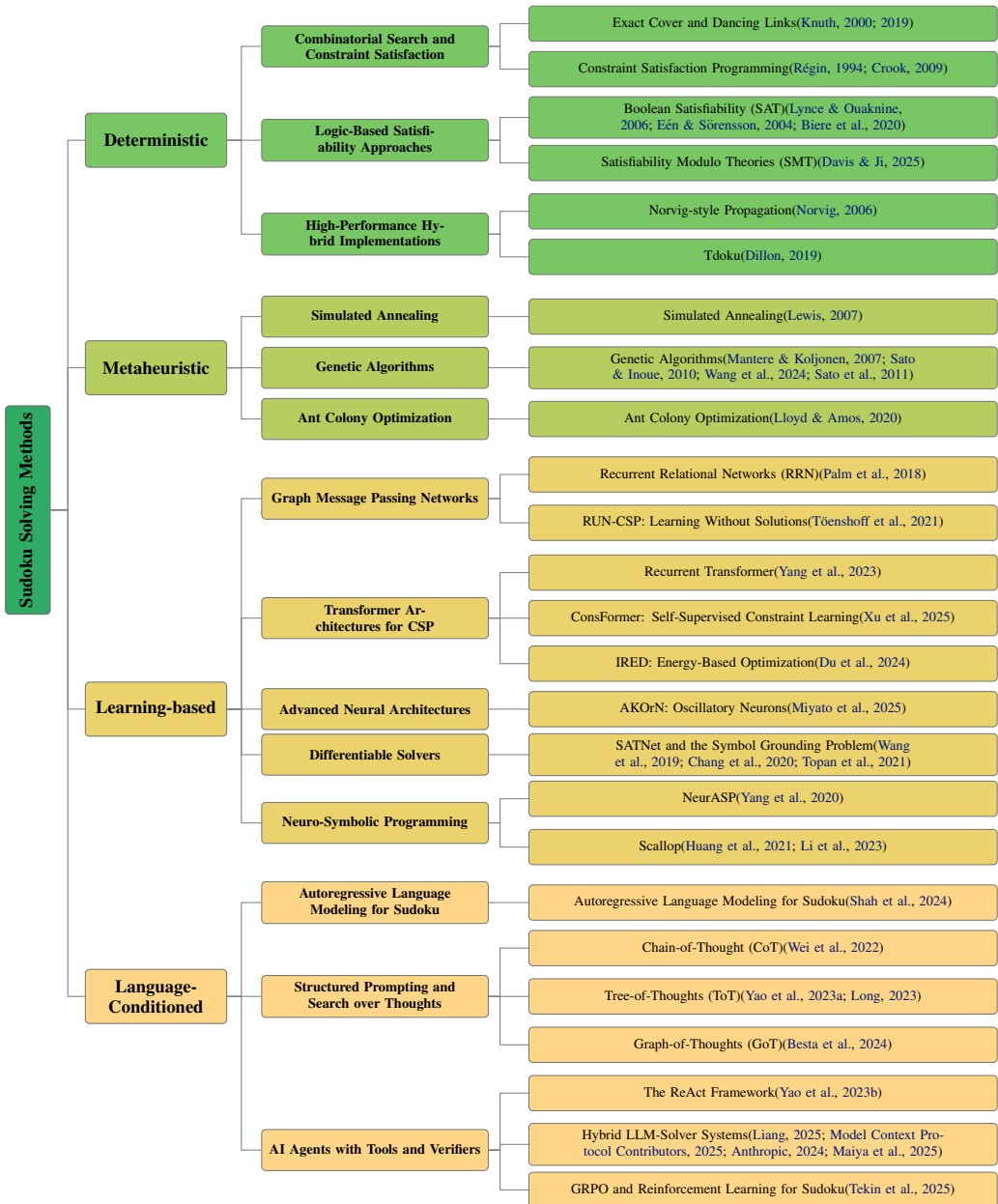

*Figure 3.* Comprehensive taxonomy of Sudoku solving methods. Four paradigms are distinguished by color: Deterministic (green) methods guarantee correctness by construction; Metaheuristic (yellow-green) methods use stochastic search; Learning-based (yellow) methods learn from data; Language-conditioned (orange) methods leverage large language models. Only deterministic methods and neuro-symbolic hybrids (which delegate to symbolic solvers) can certify constraint satisfaction.

## A. Taxonomy of Sudoku Solving Methods

Figure 3 provides a comprehensive taxonomy of Sudoku solving methods across four paradigms.

---

**Algorithm 2** Sudoku Solution Verification

---

1: **procedure** VERIFYSUDOKU(grid)
2: $n \leftarrow 9$
3: expected $\leftarrow \{1, 2, 3, 4, 5, 6, 7, 8, 9\}$
4: *// Check rows*
5: **for** $r = 0$ **to** $n - 1$ **do**
6:   **if** set(grid[$r$]) $\neq$ expected **then**
7:     **return** FALSE
8:   **end if**
9: **end for**
10: *// Check columns*
11: **for** $c = 0$ **to** $n - 1$ **do**
12:   column $\leftarrow \{\,\text{grid}[r][c] : r \in \{0, \ldots, n-1\}\,\}$
13:   **if** column $\neq$ expected **then**
14:     **return** FALSE
15:   **end if**
16: **end for**
17: *// Check 3×3 boxes*
18: **for** $br = 0$ **to** $2$ **do**
19:   **for** $bc = 0$ **to** $2$ **do**
20:     box $\leftarrow \{\,\text{grid}[3br + dr][3bc + dc] : dr, dc \in \{0, 1, 2\}\,\}$
21:     **if** box $\neq$ expected **then**
22:       **return** FALSE
23:     **end if**
24:   **end for**
25: **end for**
26: **return** TRUE
27: **end procedure**

---

## B. Verification Algorithm

All paradigms ultimately require verifying candidate solutions. The following procedure checks whether a filled 9×9 grid satisfies Sudoku constraints.

Complexity: $O(n^2)$ for an $n \times n$ grid.

## C. Complexity Summary

*Table 4.* Computational Complexity Summary (high level)

| Method | Time (Worst) | Time (Typical) | Certificate |
|---|---|---|---|
| *Deterministic* | | | |
| DLX / SAT / ASP | exponential | $\mu$s–s | Yes (assignment / proof) |
| *Metaheuristic* | | | |
| SA / GA / ACO | – | ms–s | No (needs verifier) |
| *Learning-based* | | | |
| RRN / SATNet / ConsFormer | – | ms–s | No (needs verifier) |
| *Hybrid* | | | |
| NeurASP / Tool+Solver | solver-dependent | ms–s | Yes (via symbolic) |

## D. Additional Application Domains

The position generalizes beyond Sudoku whenever constraints are specifiable and checkable:

- Drug discovery: valence rules, PAINS filters, Lipinski constraints; violations can cost \$1–2.6B per failed candidate (Di-Masi et al., 2016; Wouters et al., 2020).

- Flight scheduling: duty limits, rest requirements, maintenance windows; violations can trigger large passenger disruption costs (United States Government Accountability Office, 2011).

- Circuit design: timing, power, and design-rule checks; violations can cost millions per respin.

## E. Theoretical Background

### E.1. Why Neural Networks Cannot Guarantee Constraint Satisfaction

**Proposition E.1** (Extrapolation Regime). *In high-dimensional input spaces, neural network predictions on test data almost always involve extrapolation beyond the training distribution (Balestriero et al., 2021).*

**Proposition E.2** (Hallucination Inevitability). *For any computable function $f$, there exist inputs where a trained model $M$ produces $M(x) \neq f(x)$ with non-negligible probability (Xu et al., 2024).*

**Implication.** Unconstrained, data-implied neural networks learn statistical regularities, not logical necessities. They satisfy constraints often, not always; guarantees arise only when constraint structure is injected by design (architectural constraints) or enforced by explicit symbolic verification/solving. Hence the need for symbolic certification when violations are costly and constraints are cheap to verify.

