# OpenReview forum: "Position: Certified Correctness in Neural Constraint Reasoning Requires Symbolic Integration"
_ICML.cc/2026/Position_Paper_Track — ICML 2026 Position Paper Track regular_

### Official Review · Reviewer_rda3 · 2026-02-26

**Significance:** 3
**Argument Clarity:** 2
**Rating:** 4
**Confidence:** 3

**Questions:**

1. Could the author please explain in detail how the ConstructFeedback mechanism in Algorithm 1 is implemented? How can a neural network effectively absorb structured symbolic diagnostic information and update it for the next iteration without retraining?

2. When comparing the sample efficiency of NeurASP and RRN, there are fundamental differences between the two in terms of input features and rule priors. Can the author provide baseline comparison results that control for prior knowledge variables to prove that the efficiency improvement truly stems from the architecture rather than human-injected rules?

3. In industrial-grade tasks where the verification process is extremely time-consuming, can the efficiency and correctness of the PVS framework still be guaranteed?

**Alternative Views Section:**

Yes

**Compliance With Llm Reviewing Policy A Conservative:**

Affirmed.

**Discussion Potential:**

3

**Final Justification:**

The response has helped clarify my comments. As my rating is already positive, I plan to leave it unchanged.

**Paper Summary:**

This position paper points out that the neural-only methods suffer severe degradation when the difficulty distribution shifts and a lack of certificates in constraint satisfaction problems. By comparing and categorizing Sudoku solving methods, a certification gap was demonstrated. To address this issue, the authors propose the Proposer-Verifier-Solver (PVS) framework, which advocates leveraging a neural proposer for efficiency, a symbolic verifier as a generator of specific constraint violations, and a symbolic solver as a guarantee of obtaining a valid solution or a proof of unsatisfiability.

**Position:**

Yes

**Position In Title:**

Yes

**Related Work:**

3

**Strengths And Weaknesses:**

## Strengths
+ The paper introduces the certification gap and proposes the PVS framework to address the pain point of performance degradation of neural solvers when the distribution changes, with a clear motivation.

+ The paper clearly states its position and provides falsifiable conditions.

+ The categorization of current solution paradigms (deterministic algorithms, metaheuristic optimization, end-to-end neural learning, and language-conditioned reasoning) has certain structured reference value.


## Weaknesses
- The architectural innovation in the paper lacks technical depth. The PVS framework is essentially a replica of the classic Guess-and-Check approach or Counterexample-Guided Abstraction Refinement in formal verification. In algorithm 1, the method for converting diagnostics into prompt, i.e., the ConstructFeedback function, is not explained, making the overall framework superficial.

- The experimental comparison also contains a logical fallacy. In demonstrating sample efficiency (Claim 3), NeurASP, which has built-in complete constraint rules as prior knowledge, is compared to RRN, which must learn rules from scratch. This comparison of uncontrolled variables is unfair and cannot scientifically prove its claims.

- The paper oversimplifies the complexity of the industrial scenario.. The paper's argument heavily relies on the asymmetric assumption that "verification is extremely fast (polynomial time), while solving is extremely slow." However, in complex CSPs such as real-world software program verification, the verification process itself often requires calling heavyweight SMT solvers, which is extremely costly. The paper lacks empirical support for scenarios with higher verification complexity, outside of Sudoku datasets.

**Support:**

2

---

> ### Author Rebuttal · Authors · 2026-03-31
>
> We thank Reviewer rda3 for the most technically incisive review and address all three weaknesses below.
>
> **W1 / Q1: ConstructFeedback; PVS vs. CEGAR.**
>
> **Classical guess-and-check** discards candidates and restarts with no information. **CEGAR** (Clarke et al., CAV 2000) returns a counterexample trace over the global abstract model's state space and uses it for a *system-level* operation: refining the global abstraction by adding predicates, affecting all subsequent model-checking runs indiscriminately. Neither delivers spatially localized feedback about the specific violated sub-region of a candidate.
>
> **PVS differs at two levels.** First, verifier output is spatially localized, not a global trace:
>
> | Setting | Verifier output |
> |---|---|
> | Sudoku | "Cells (3,4) and (3,7) share value 5; row-3 all-different violated" |
> | Code generation | File path, line, column, AST error node, violation type |
> | SMT-based verification | UNSAT core: minimal conflicting clause set with literal assignments |
>
> Second, `ConstructFeedback` injects this into the neural proposer's representation space for the next forward pass, without any parameter update:
>
> - **LLMs**: diagnostics prepended as execution context; in-context learning reshapes attention (Brown et al., NeurIPS 2020). LLMs demonstrably fix outputs from verifier feedback without fine-tuning (Chen et al., ICLR 2024; Olausson et al., ICLR 2024).
> - **GNNs**: diagnostics → edge masks / penalty flags on violating nodes; next forward pass conditioned on modified graph structure.
> - **Diffusion/EBMs**: diagnostics → ∇_y E_sym(y) steering sampling at inference (Du & Mordatch, NeurIPS 2019; Ye et al., NeurIPS 2024). AKOrN in Section 4.2 is precisely this realization, already demonstrated in the paper's own experiments.
>
> Parameters never change; only the input/conditioning changes (standard test-time adaptation). The result is **iterative refinement**, not restart: fallback rate decreases across iterations (CaR: 38.22% → 0.03% over 10 steps).
>
> **W2 / Q2: NeurASP-RRN comparison; controlled baseline.**
>
> Claim 3 is a **paradigm-cost measurement**, not a learning-algorithm comparison. The independent variable is the presence vs. absence of symbolic specification. The reviewer asks for a "controlled baseline without symbolic rules": this **already exists in the data**: RRN is NeurASP without symbolic rules, trained on 180,000 samples (7,200× more than NeurASP's 25 perception training samples). Yet RRN still collapses on the identical OOD shift.
>
> This is not a sample-efficiency failure. No amount of data gives a soft-regularizer architecture the property-level guarantee that a symbolic verifier provides; gradient descent on discrete constraints produces soft approximations that degrade under distribution shift by construction. The certification gap is architectural, not empirical. We will add a clarifying paragraph at Section 3.3 making this design intent explicit.
>
> **W3 / Q3: High-cost industrial verification.**
>
> Our position rests on **relative asymmetry**, not absolute cheapness: for all NP problems, verification is asymptotically cheaper than exhaustive solving by definition of the complexity class. Even SMT-based verification checks a partially-optimized neural candidate, which is typically much cheaper than solving from scratch.
>
> Industrial practice confirms multi-tiered staged verification reduces amortized cost (Chong et al., "Systems Correctness Practices at Amazon Web Services," ACM Queue 2024; Newcombe et al., CACM 2015). Neural candidates are filtered in successive passes: (1) O(n) syntax parsing, rejecting syntactically invalid candidates at negligible cost; (2) lightweight type checking and constraint propagation, filtering structurally valid but semantically incorrect candidates before any SMT call; (3) bounded model checking, applied only to candidates passing prior filters; (4) full SMT, invoked only for the small fraction passing all prior filters. In our Sudoku experiment, 89.5% of proposals pass the O(n²) checker before reaching the symbolic solver; amortized cost is 0.895 × O(n²) + 0.105 × O(solver), not O(solver) per query. If 80–90% of LLM-generated programs are eliminated by syntax and type checks, amortized SMT cost is 10–20% of running SMT on every candidate.
>
> When full verification is expensive: (i) timeouts → fallback to safe template (no unverifiable output emitted); (ii) partial verification certifies the tractable sub-specification, strictly safer than zero certification from pure neural; (iii) UNSAT core provides actionable diagnostics unavailable from any pure neural approach. PVS's scope conditions (Section 1, criteria i–iii) already limit efficiency claims to the asymmetric regime.
>
> **Camera-ready**: `ConstructFeedback` pseudocode + three realizations + CEGAR comparison; Section 3.3 clarifying paragraph; Section 6.3 expansion with tiered verification, amortized cost, and partial certification (Chong et al., ACM Queue 2024).

---

> > ### Author Rebuttal · Reviewer_rda3 · 2026-04-01
> >
> > I thank the authors for the responses! Since my score is already positive, I decide to remain my score unchanged.

---

### Official Review · Reviewer_dhMZ · 2026-03-11

**Significance:** 4
**Argument Clarity:** 4
**Rating:** 5
**Confidence:** 4

**Questions:**

My questions relate to the listed weaknesses:
 1. Are there other case studies besides Sudoku that demonstrate a similar shortcomming of neural-only systems?
 2. Are there tools other than NeurASP that combine a neural and and a symbolic approach, where the neural network is actually part of the generation of candidate solutions? If so, how do they perform on the OOD data? How often does the fallback symbolic solver (if present) need to be called?

**Alternative Views Section:**

Yes

**Compliance With Llm Reviewing Policy A Conservative:**

Affirmed.

**Discussion Potential:**

3

**Final Justification:**

The authors have addressed my questions and concerns, particularly by expanding on the demonstrated applications.

**Paper Summary:**

The paper argues that in domains where finding a solution to a given problem is expensive, but verifying its correctness is relatively cheap (such as Sudoku), neural solutions alone are insufficient. The authors refer to results that show that such implementations fail with high error rates on OOD inputs. The authors argue that instead, the research community should focus on combinations of neural "proposers" and symbolic "verifiers", where the verifier can provide a sound evaluation of the provided solution candidates. If the solution is invalid, it should provide instructive feedback to the neural network (e.g. by pinpointing the violated rule) instead of just returning a binary result.

The authors make three claims:
1. Neural-only systems will have an error rate > 10% on OOD data
1. Simply scaling test-time compute will not solve this
1. Integrated systems are more efficient by multiple orders of magnitude

Specifically the first one (in a strengthened form, requiring an error rate < 1% on OOD data) is presented as a refutable statement.

**Position:**

Yes

**Position In Title:**

Yes

**Related Work:**

3

**Strengths And Weaknesses:**

# Strengths
The authors state a clear position which is relevant to the ICML community: At a time where neural networks become increasingly powerful, often by means of scaling them even further, highlighting the importance of incorporating other (formal) tools seems very relevant.

The papers "Opposing Views" section provides steelman arguments for each view, making the argumentation for and against their position clear and easy to follow.

In general, the paper is well written and easy to read.



# Weaknesses
The failure of neural-only architectures is empirically proven via the example of Sudoku. While the authors address the potential concern that this does not translate to real-world use cases, having more than one example in the paper would be beneficial to demonstrate the generality of the statement.

Giving NeurASP as an example for a combination of a neural and a formal approach is valid, but extreme: Based on the authors description, this tool only uses a neural network to process the input data. Actually solving the Sudoku is completely delegated to a symbolic solver.

**Support:**

3

---

> ### Author Rebuttal · Authors · 2026-03-31
>
> We thank Reviewer dhMZ and we address both weaknesses and both questions below.
>
> **W1 / Q1: Additional case studies: neural-only shortcomings and the propose-verify pattern across domains.**
>
> Three domains demonstrate (a) neural-only failure under hard constraints and (b) propose-verify-refine improvement with explicit fallback trigger rates:
>
> **Code generation** (verifier: compiler/static analyzer): ICF agents instantiate the propose (LLM) / verify (compiler) / refine (diagnostics) loop and achieve 91% accuracy on ArkTS/HarmonyOS (Aytekin et al., AutoSE 2026). The compiler returns file/line/column/type diagnostics that condition the LLM's next generation without weight updates. ~9% require a safe template fallback.
>
> **Hard vehicle routing, TSPTW-100** (verifier: constraint feasibility checker): AM/POMO neural solvers exhibit 38.22% infeasibility on TSPTW-100 (Table 2, Bi et al., ICLR 2026). PIP (Bi et al., NeurIPS 2024) reduces this to 6.96%; CaR (Bi et al., ICLR 2026) drives it to 0.03% within T_R=10 steps. Fallback rate starts at ~38% and converges to near-zero.
>
> **Automated theorem proving, miniF2F** (verifier: Lean proof kernel): Best single-pass neural prover (Kimina-Prover, Wang et al., arXiv:2504.11354, 2025): ~53% pass@1. BFS-Prover-V2 (Xin et al., arXiv:2509.06493, 2025) with Lean as mandatory verifier at every BFS node: 95.08%. Failed steps trigger backtrack; ~5% remain unsolved after exhausting the budget.
>
> Summary:
>
> | Domain | Neural-only failure | Fallback rate | Final accuracy |
> |---|---|---|---|
> | Code gen, ArkTS (ICF) | Substantial syntax errors | ~9% → safe template | ~91% |
> | Hard VRP, TSPTW-100 (CaR) | 38.22% infeasible (POMO) | ~38% → refinement → 0.03% | 99.97% |
> | Theorem proving, miniF2F (BFS-Prover-V2) | ~47% fail (best ~53% single-pass) | failed steps → BFS backtrack | 95.08% |
>
> **W2 / Q2: Two paradigms beyond NeurASP; OOD performance; fallback rates.**
>
> NeurASP is at one extreme: its neural component performs only perception (mapping images to digit distributions); the ASP solver constructs the full solution entirely. The neural network does not generate candidate solutions. Beyond NeurASP, two distinct paradigms exist where the neural network genuinely participates in generating candidate solutions; they differ fundamentally in how the symbolic component is integrated:
>
> **Paradigm A: Differentiable soft symbolic** (SATNet, Scallop): symbolic constraints embedded as differentiable approximations in the training pipeline. SATNet (Wang et al., ICML 2019) relaxes MAXSAT to a semidefinite program (SDP) for end-to-end training; Scallop (Huang et al., NeurIPS 2021) approximates logical inference via weighted provenance semirings. The symbolic structure is approximated, not enforced; softened constraints hold only in expectation and degrade under distribution shift. OOD performance: SATNet **98.3% in-distribution → 3.2% OOD** (Miyato et al., ICLR 2025, Table 3). No fallback; silent failure.
>
> **Paradigm B: Hard symbolic verification** (ICF, PIP/CaR, BFS-Prover-V2, PVS): the neural network generates complete candidates; a hard symbolic verifier checks each and returns structured diagnostics for the next refinement. The verifier cannot be bypassed. OOD: 91%–99.97%–95.08% with measurable, converging fallback rates (table above).
>
> Our **controlled experiment (Section 4.2)** provides exact fallback counts: 89.5% of AKOrN's candidates pass the O(n²) constraint checker (<1 µs per puzzle); 10.5% trigger the symbolic fallback solver and are certified correct.
>
> | Paradigm | Tool | Neural-only OOD failure | Fallback rate | Final certified accuracy |
> |---|---|---|---|---|
> | A: Soft symbolic (no fallback) | SATNet (Sudoku OOD) | 96.8% error | none (silent failure) | 3.2% |
> | B: Hard symbolic verifier | AKOrN-PVS (Sudoku OOD, our paper) | 10.5% before fallback | 10.5% to symbolic solver | 100% |
> | B: Hard symbolic verifier | ICF (code gen, ArkTS) | substantially above 9% | ~9% to safe template | ~91% |
> | B: Hard symbolic verifier | PIP/CaR (hard VRP, TSPTW-100) | 38.22% (POMO baseline) | ~38%, converges to 0.03% | 99.97% |
> | B: Hard symbolic verifier | BFS-Prover-V2 (miniF2F) | ~47% (best ~53% single-pass) | failed steps → BFS backtrack | 95.08% |
>
> The 3.2% vs. 100% contrast on the **same Sudoku benchmark** is the paper's central empirical argument.
>
> **Camera-ready**: Section 5.4 with cross-domain summary table, explicit fallback trigger rates, and neural-only failure rates; Section 5 paragraph explicitly distinguishing NeurASP / Paradigm A / Paradigm B (PVS); revise Section 4.2 NeurASP discussion to clarify its control-baseline role.

---

> > ### Author Rebuttal · Reviewer_dhMZ · 2026-04-07
> >
> > Thank you for your rebuttal and addressing my questions! I think adding this feedback might improve the paper. I stand by my original rating.

---

### Official Review · Reviewer_Xvrb · 2026-03-13

**Significance:** 3
**Argument Clarity:** 3
**Rating:** 5
**Confidence:** 1

**Questions:**

- Line 085, left column, scope note. What about the problems that architectural enforcement is easy? In this case, neural systems should be comparable to explicit symbolic verification? Or do all the systems tackling combinatorial problems with architectural enforcement have this issue? I would like to see some discussions on this.
- Line 166, left column, >10%. This is a very specific number that should be problem-dependent or at least dependent on how far the distribution shifts. Why choosing this number?
- Line 318, right column, 'allowing the neural model to correct specific errors without retraining'. Is it related to specific class of models? In Algorithm 1, the authors use the term prompt, and $\theta$ is not modified. Are there only a special kind of models such as LLMs are applicable in your algorithm?
- Is Sudoku a typical combinatorial problem with polynomial verification? Can you discuss more NP-complete problems with different difficulty levels of verification?

**Alternative Views Section:**

Yes

**Compliance With Llm Reviewing Policy A Conservative:**

Affirmed.

**Discussion Potential:**

3

**Final Justification:**

After reading authors' rebuttal and other reviewers' comments, I believe the novelty of the proposed position is fine and of relevance to the community. However, I suspect the rebuttal is LLM-generated. If not, I will vote for accept. I have lowered my confidence because of this.

**Paper Summary:**

This paper argues with a position that neural constraint reasoning should integrate and prioritize symbolic verification in settings with hard constraints, easy verification, and high violation cost. The authors introduce four constraint satisfaction paradigms, three empirical claims on neural-only and neural-symbolic methods. The authors propose a Proposer-Verifier-Solver (PVS) framework provably correct methods and provide falsifiability for refuting the proposed position.

**Position:**

Yes

**Position In Title:**

Yes

**Related Work:**

3

**Strengths And Weaknesses:**

**Strengths**
- This paper presents falsifiability, which sounds reasonable, and is a rare case based on my assigned bunch of papers to be reviewed.
- The paper is well written, with few typos, and easy to follow.
- Empirical evidence and theoretical ground from related works make sense for their position.

**Weaknesses**
- Only one problem instance, Sudoku, is studied, whose representativeness is questionable.

**Support:**

4

---

> ### Author Rebuttal · Authors · 2026-03-31
>
> We thank Reviewer Xvrb and address the stated weakness and all four questions.
>
> **Weakness: Sudoku representativeness.** Sudoku is a controlled **"Drosophila model"**: by varying only the clue count from in-distribution [31-42] to out-of-distribution [17-34] while holding all 324 constraints fixed, we isolate OOD failure attributable purely to neural approximation. Three high-stakes domains confirm generalization:
>
> - **Code generation** (verifier: compiler/static analyzer): LLM generates code; compiler returns file/line/column/type diagnostics; LLM refines zero-shot. ICF: 91% accuracy on ArkTS/HarmonyOS (Aytekin et al., AutoSE 2026); ~9% require safe template fallback.
> - **Hard VRP, TSPTW-100** (verifier: constraint checker): AM/POMO neural solvers exhibit 38.22% infeasibility (Bi et al., ICLR 2026, Table 2). PIP (NeurIPS 2024): 6.96%; CaR (ICLR 2026): 0.03% within T_R=10 steps. Fallback rate ~38% converges to near-zero.
> - **Theorem proving, miniF2F** (verifier: Lean kernel): BFS-Prover-V2 (Xin et al., arXiv:2509.06493, 2025): 95.08%; best single-pass neural (Kimina-Prover, Wang et al. 2025): ~53% pass@1.
>
> These three domains span code, combinatorial optimization, and formal mathematics, confirming the propose-verify-fallback pattern generalizes beyond Sudoku. We will add Section 5.4 consolidating these with a summary table.
>
> **Q1 (Line 085): Architectural enforcement.** For constraints expressible as continuous functions, architectural enforcement works and symbolic verification adds no value; examples: probability simplexes via Softmax, linear inequalities via differentiable quadratic programs (OptNet, Amos & Kolter, ICML 2017). PVS targets the *complementary* regime: **discrete/combinatorial constraints** with global structure (all-different in Sudoku, subtour-elimination in TSP, chromatic constraints in graph coloring). For these, neural solvers necessarily rely on continuous relaxations during training (SATNet uses a differentiable SDP relaxation of MAXSAT), providing effective soft guidance in-distribution but unable to guarantee exact discrete satisfaction at inference. The discrete rounding step reintroduces violation risk by construction, and this cannot be resolved by network architecture alone without resolving P vs NP (Garey & Johnson, 1979). Architectural and symbolic enforcement are *complementary*, not competing.
>
> **Q2 (Line 166): Why >10%?** The 10.5% is the board-error rate of AKOrN (Miyato et al., ICLR 2025), the current state-of-the-art neural-only Sudoku solver, at its **maximum test-time compute**: 128 Kuramoto synchronization steps, 4,096 random initial oscillator samples, energy-based voting. This is hard to be reduced within the neural-only paradigm: without the full budget, OOD accuracy is only ~ 18% (~ 82% error); with 128 steps but no voting, ~ 51% (~ 49% error); with the full budget, 89.5% ( ~ 10.5% error). Further compute scaling cannot close this gap. Claim 1 will be revised to specify: " ~ 10.5% for AKOrN at maximum test-time compute under the defined Sudoku OOD protocol."
>
> **Q3 (Line 318): ConstructFeedback; LLMs only?** PVS is architecture-agnostic; `prompt` was imprecise shorthand. `ConstructFeedback` implements **Cross-Space Test-Time Conditioning**: structured symbolic diagnostics injected into the proposer's representation space without any parameter update:
> - **LLMs**: diagnostics prepended as execution context; in-context learning (Brown et al., NeurIPS 2020) reshapes attention. LLMs demonstrably fix outputs from verifier feedback without fine-tuning (Chen et al., ICLR 2024; Olausson et al., ICLR 2024).
> - **GNNs**: diagnostics → edge masks / penalty flags on violating nodes; conditions next forward pass without retraining.
> - **Diffusion/EBMs**: diagnostics → ∇_y E_sym(y) steering sampling at inference (Du & Mordatch, NeurIPS 2019). **AKOrN in Section 4.2 is an energy-based model; this realization is already demonstrated in the paper's own experiments.**
>
> **Q4: Verification complexity across NP.** Sudoku's O(n²) verifier is *heavier* than most NP-complete problems:
>
> | Problem | Verifier | Relative to Sudoku |
> |---|---|---|
> | 3-SAT | O(n) (per-clause check) | Cheaper |
> | Hamiltonian Cycle | O(n) (path validity) | Cheaper |
> | Graph k-Coloring | O(E) (edge conflict) | Cheaper |
> | Sudoku | O(n²) (row/col/box) | Baseline (conservative) |
>
> PVS advantages strengthen on lighter-verification NP problems. Beyond NP (PSPACE/EXPTIME), PVS retains value via cheap falsification and bounded partial verification, strictly safer than zero certification from pure neural.
>
> **Camera-ready**: Section 5.4 cross-domain summary; rename `prompt` to `conditioned_context` + `ConstructFeedback` pseudocode; verification-complexity table; beyond-NP discussion in Section 6.3; complementarity statement in Scope note.

---

> > ### Author Rebuttal · Reviewer_Xvrb · 2026-04-01
> >
> > Thanks for your response. I will keep my score positive.

---

### Official Review · Reviewer_BCdy · 2026-03-19

**Significance:** 3
**Argument Clarity:** 3
**Rating:** 4
**Confidence:** 4

**Questions:**

Can the authors discuss whether this framework extends to modern applications such as AI for code generation, where a neural model parses the user’s intent from natural language and a symbolic verifier checks the generated code for bugs or formal constraint violations? In this regard, Sudoku may be too narrow as a motivating example to fully demonstrate the broader practical importance of this research direction.

**Alternative Views Section:**

Yes

**Compliance With Llm Reviewing Policy A Conservative:**

Affirmed.

**Discussion Potential:**

3

**Final Justification:**

My concerns have been adequately addressed.

**Paper Summary:**

This paper suggests that neural constraint reasoning should incorporate symbolic verification into learning based models, especially for those applications with hard constraints, cheap verification, and high costs of failure.

Using Sudoku as the main example, it surveys many different approaches and proposes a main framework in which neural models generate candidates and symbolic components verify or solve them.

**Position:**

Yes

**Position In Title:**

Yes

**Related Work:**

2

**Strengths And Weaknesses:**

Strength

The paper pitched an important and timely question: whether high empirical accuracy in neural reasoning is sufficient for domains with strict logical requirements.

Weakness
2. The scope may be too narrow, since many real-world problems do not come with clean symbolic solvers or fully specified hard constraints, so the proposed position may not generalize as broadly as suggested.
3. The proposed message is somewhat incremental, because integrating neural models with symbolic solvers or verifiers has already been explored extensively in prior neuro-symbolic and tool-augmented reasoning work, rather than being a fundamentally new perspective.

**Support:**

3

---

> ### Author Rebuttal · Authors · 2026-03-31
>
> We thank Reviewer BCdy and address W2, W3, and the question below.
>
> **W2: Scope.**
>
> **(a) "Many real-world problems lack clean symbolic solvers."** PVS requires only a lightweight **verifier** (checker), not a complete **solver**. These are architecturally distinct: a verifier only needs to check whether a given candidate satisfies the constraints; it does not need to find a solution from scratch. Compilers check type and syntax constraints without re-solving type inference. Scheduling systems check resource conflicts without constructing schedules. Lean/Coq kernels validate individual proof steps without discovering proofs. None require a "clean symbolic solver" in the traditional sense; they are lightweight checkers that exist in virtually every constrained engineering domain. When a fallback solver is unavailable, a safe template or graceful exception suffices. PVS guarantees no unverifiable output reaches the user.
>
> **(b) "Constraints are often partially specified."** Section 6.3 addresses this in three sub-cases: (i) **Partially-known constraints**: verifier certifies all checkable hard constraints; partial certification is strictly safer than zero. (ii) **Soft constraints**: exact penalty values rather than a binary signal, for principled conditioning. (iii) **Infeasible instances**: UNSAT with Minimal Unsatisfiable Core, enabling targeted constraint relaxation. PVS degrades gracefully in all cases. The paper's position is scoped to domains satisfying at least a partial verifier (criteria i–iii, Section 1).
>
> Three deployed domains confirm generality:
>
> | Domain | Neural-only failure | With propose-verify loop |
> |---|---|---|
> | Code gen, ArkTS (ICF, Aytekin et al., AutoSE 2026) | Substantial syntax errors | 91% accuracy; ~9% fallback to safe template |
> | Hard VRP, TSPTW-100 (PIP/CaR, Bi et al., NeurIPS 2024/ICLR 2026) | 38.22% infeasible (POMO) | 0.03% (CaR, T_R=10) |
> | Theorem proving, miniF2F (BFS-Prover-V2, Xin et al., arXiv:2509.06493) | ~47% fail (best ~53%, Kimina-Prover, Wang et al. 2025) | 95.08% |
>
> **W3: Incrementality.** The novel contribution is **Inversion of Control** at the architecture level, not the concept of combining neural and symbolic. Two prior paradigms fail structurally:
>
> *(A) Differentiable soft symbolic* (SATNet, Scallop): constraints as soft regularizers; OOD failure is structurally inevitable since softened discrete constraints hold only in expectation. SATNet: **98.3% in-distribution → 3.2% OOD** (Miyato et al., ICLR 2025, Table 3). Silent failure; no fallback.
>
> *(B) Ad-hoc tool-use* (LLM-Modulo, Toolformer, ReAct): neural retains full control over when to invoke the tool and whether to accept its output. Kambhampati et al. (ICML 2024) document two systematic failure modes: *invocation hallucination* (the model generates an answer instead of calling the tool) and *output override* (the model discards the correct symbolic result because it conflicts with the parametric prior). Correctness remains probabilistic even when a perfect verifier is available.
>
> **PVS** enforces an Inversion of Control that neither paradigm achieves. The symbolic verifier is the mandatory gatekeeper at the system level; the neural proposer is architecturally precluded from bypassing or overriding it. The Certification Invariant (Proposition 5.1) and formal proof in Section 5.3 guarantee every execution path of Algorithm 1 exits only through a verified output (a structural guarantee, not a probabilistic bound). This is the qualitative leap from "usually correct" to *provably zero-violation*.
>
> **Question: Code generation; is Sudoku too narrow?**
>
> | PVS component | Code generation instantiation |
> |---|---|
> | P_θ (proposer) | LLM → candidate code from natural-language intent |
> | V (verifier) | Compiler / static analyzer → file/line/column/type diagnostics |
> | D (diagnostics) | Structured errors enabling zero-shot LLM refinement without retraining |
> | S (fallback) | Program synthesis engine or safe template |
>
> Sudoku is a controlled **"Drosophila model"**: varying only the clue count ([31-42]→[17-34]) while holding all constraints fixed induces a clean OOD shift attributable to neural approximation alone; methodological control difficult to achieve in code generation where many confounders change simultaneously. The cross-domain table above demonstrates real-world generalization.
>
> **Camera-ready**: three-way paradigm comparison table (A/B/PVS) in Related Work with 2024-2025 references; Section 5.4 cross-domain summary with explicit fallback trigger rates; Section 6.3 graceful-degradation expansion for partial/soft/infeasible constraint cases; explicit Sudoku-as-controlled-testbed framing in Section 1 to preempt scope misreading.

---

> > ### Author Rebuttal · Reviewer_BCdy · 2026-04-03
> >
> > My concerns have been adequately addressed.

---

### Decision · Program_Chairs · 2026-04-30

**Decision:**

Accept (regular)

**Comment:**

This paper discusses settings in which symbolic verification should be integrated into neural reasoning problems with strict constraints. The position is clear and timely. The position is argued rhetorically, and also using a compilation of empirical studies on solving Sudoku. These arguments are somewhat weak, but in the rebuttals the authors have promised to include additional studies on other domains.

One issue that is ignored here is autoformalization, which would be needed in many interesting problems using LLMs as proposers. For example, in automated theorem proving using LLMs, one can either translate a proposed solution to Lean and verify it, or try to verify directly using a model. It's unclear that autoformalization is easier than verification for models trained using statistical methods. Maybe settings requiring autoformalization are outside of the scope of the paper, but in that case the scope is fairly limited.

Given the positive reviews, and given that symbolic solvers are not standard practice, I am in favor of accepting this paper. In the final version, the authors should expand the evidence to include non-Sudoku domains, and discuss autoformalization / neural verifiers in the Alternative Views.